# Yield and Nutritional Response of Greenhouse Grown Tomato Cultivars to Sustainable Fertilization and Irrigation Management

**DOI:** 10.3390/plants9081053

**Published:** 2020-08-18

**Authors:** Vasile Stoleru, Simona-Carmen Inculet, Gabriela Mihalache, Alexandru Cojocaru, Gabriel-Ciprian Teliban, Gianluca Caruso

**Affiliations:** 1Department of Horticultural Technologies, “Ion Ionescu de la Brad” University of Agricultural Sciences and Veterinary Medicine, 3 M. Sadoveanu, 700440 Iasi, Romania; gabriela.mihalache.gm@gmail.com (G.M.); cojocaru.alexandru@yahoo.com (A.C.); gabrielteliban@uaiasi.ro (G.-C.T.); 2Integrated Center of Environmental Science Studies in the North East Region (CERNESIM), The “Alexandru Ioan Cuza” University of Iasi, 700449 Iasi, Romania; 3Department of Agricultural Sciences, University of Naples Federico II, Naples, 80055 Portici, Italy; gcaruso@unina.it

**Keywords:** *Solanum lycopersicum* L., organic fertilization, irrigation regime, production, antioxidants, mineral elements

## Abstract

Tomato is considered one of the most important crops worldwide from nutritional and economic standpoints, and, in this respect, sustainable production should be a prime objective, particularly in terms of fertilization and irrigation management. The aim of this study was to compare the effects of two fertilization types (chemical or organic) and two irrigation regimes (67% or 100% of evapotranspiration replenishment) on biometrical, biochemical, and yield parameters of three indeterminate cultivars of tomato grown in a greenhouse. The results showed that the effect of organic fertilization was better compared to chemical fertilization for lycopene accumulation and antioxidant activity, as well as for the lower concentrations of any of the macroelements in the tomato fruits; therefore, organic fertilization can be used as an alternative to chemical fertilization in sustainable horticulture. In each cultivar under the same fertilization type, the effect of irrigation was significant on yield and the number of fruits, but the 100% evapotranspiration restoration did not enhance the fruit concentration of all the macroelements and microelements compared to 67% irrigation regime. Higher concentrations of macro- and microelements in the chemically fertilized fruits compared to the organic ones, regardless of the cultivar and the irrigation regime, suggest that the inorganic substances are more easily absorbed by plants under a protected environment. Organic fertilization positively affected the lycopene and antioxidant activities of tomato fruits, thus proving to be a valuable alternative to chemical fertilization in sustainable agriculture, although the product premium quality also depends on the cultivar used.

## 1. Introduction

Tomato (*Solanum lycopersicum* L.) is one of the main cultivated vegetables in the world [1], and, in Romania, it is grown as forced and protected crops on about 50,000 ha, which is equivalent to 60–65% of the cultivable area in greenhouses [2]. The economic importance of this crop is due to the staggered use of labor as a result of the different systems and forms of cultivation, the high yields, and the resulting income [3]. The regular consumption of fresh and processed tomatoes has been associated with a low incidence of various types of cancer, considering the high content of antioxidant compounds [4]. Indeed, the added nutritional value of tomato fruits is represented by secondary metabolites with antioxidant properties, such as lycopene, α- and β-carotene, violaxanthin, neoxanthin, lutein, zeaxanthin, α- and β-cryptoxanthin, neurosporene, phytoene, phytofluene, cyclolopene, p-coumaric acid, and rutoside. α- and β-carotene, as well as β-cryptoxanthin, are important sources of vitamin A, known to increase both the humoral and cellular responses [5], whereas lycopene is a carotenoid that reduces the risk of chronic diseases such as cancer and cardiovascular disease [4]. The antioxidant activity of tomato fruits depends on the genotype, but also on the maturation stage, cultivation practices (water availability, mineral nutrients), and climate (mostly light and temperature) [6]. Various studies have shown values of lycopene content in tomato fruits ranging from 1.66 to 39.7 mg·g^−1^d.w. (dry weight), depending on the cultivars used, but in the latest high-lycopene (HLY) lines, the level of lycopene can be significantly higher [7]. The antioxidant accumulation in the fruits is also affected by the crop system. Nutrition management plays a major role [5,6], and indeed, specialized production is oriented to meet the increased consumer demands for organic products, characterized by reduced use of chemical inputs [8].

Previous research on tomato [9] showed no significant differences between organic and conventional management on yield components, as well as on plant biomass and leaf area. The organic fruits had a higher concentration of sulfur, organic acids, and vitamin C, as well as higher lipophilic antioxidant activity. However, no significant differences were recorded for the polyphenols and the hydrophilic antioxidant activity, whereas a higher lycopene concentration was detected in the conventional fruits. In other vegetable crops such as asparagus (*Asparagus officinalis*) [10], conventional management led to higher yields, whereas organic management increased the mean weight per shoot, as well as the sugar concentration of the shoots. Similarly, red basil (*Ocimum basilicum “Purpurascens”*) showed higher yield under chemical management and enhanced anthocyanins concentration and antioxidant activity in the organically grown leaves [11]. Chemical fertilization is still one of the major agricultural practices to achieve high yields, but organic fertilization represents a viable alternative to obtain high tomato fruit quality, mineral composition, and taste, which are also affected by cultivar [12,13]. Investigations have been carried out on the effects of the aforementioned factors on the nutritional and organoleptic characteristics of tomato fruit [9,14], as well as related species such as eggplant [15,16,17] and early potato [18]. Open field research has demonstrated the significant effects of unconventional fertilization (biologic or organic) on the nutrient composition of tomato fruits [8,12]. Many studies have highlighted the remarkable concentration of minerals such as K, P, and Mg in tomato fruits [9,13,19], which are essential for the human organism. In this respect, the consumption of one portion of 200 g tomatoes represents 11%, 7.5%, and 5% of the recommended daily dose for adults of K, P, and Mg, respectively.

The aim of this study was to compare the effects of chemical and organic fertilization, in factorial combination with 67% and 100% evapotranspiration (ET) replenishment, on yield, growth, antioxidants, and elemental composition of tomato fruits of three cultivars grown in a greenhouse in the northeastern area of Romania.

## 2. Results and Discussion

### 2.1. Biometrical and Yield Parameters

As there were no interactions between the year of research and the three experimental factors applied for any measured parameters, all the values are reported as means of the two investigation years (Table 1).

Fertilization type or irrigation regime significantly influenced the tomato plant height, contrasting with the reports of other authors [20]. However, in terms of yield, cultivar HTP F_1_ showed a 29.5% higher value than IB, owing to a higher (10.5%) number of fruits per plant and a higher mean weight per fruit.

The fertilization type did not significantly influence either the tomato plant height or yield components. The lack of significant fertilization effects may be attributed to the fact that the organic fertilizers release the nutrients slowly, compared to chemical ones, and, indeed, organic fertilization resulted in a lower number of fruits that reached a higher individual weight, although the difference was not significant [21]. In previous research [22], processing tomato cultivated in an organic cropping system showed reductions of leaf area (−36%), which led to lower fruit yield and total biomass dry weight (−25%). 

Regarding the influence of the two irrigation regimes, statistically significant differences were recorded only for the fruit number per plant and the yield, which generally caused a decline in fruit weight. Indeed, the irrigation with 300 m^3^·ha^−1^ resulted in a 19.3% higher yield compared to the 200 m^3^·ha^−1^ irrigation regime. The results highlight the fact that when plants are grown in protected areas, the 100% replenishment of evapotranspiration is more favorable to yield as compared to 67% replenishment, presumably due to the fulfillment of the plants’ water requirement [23]. Other authors [24] have reported that the reduction of 40% and 50% in leaf water potential generally caused a fruit weight drop, depending on cultivar and cluster. The water deficit treatment also affected the organoleptic characteristics of cherry tomato fruits and generally resulted in increased carotenoid concentration and decreased polyphenols, depending on the cultivar.

The results of the interactions between the three experimental factors applied (cultivar, fertilization type, irrigation regime) on the plant height, number of fruits per plant, mean fruit weight and yield are presented in Table 2.

No significant interactions between the experimental factors applied arose for tomato plant height (Table 2).

The fruit number per plant for Siriana F_1_ cultivar, chemically fertilized and watered with 300 m^3^·ha^−1^, showed a significant increase of 46.4% as compared to the organically fertilized plants irrigated with 200 m^3^·ha^−1^. Significant differences were also recorded between S Ch 300 and IB Ch 200, with a significantly lower number of fruits for cultivar IB, which was chemically fertilized and irrigated according to 67% ET.

Additionally, a tendency was observed for most of the plants treated with organic fertilizer, regardless of the water replenishment percentage, to have a lower fruit number per plant, though with no statistical significance, suggesting that chemical fertilization elicits a higher number of fruits. 

The mean weight per fruit significantly increased by 41.2% from the experiment treatments S Ch and IB Og, both irrigated with 200 m^3^·ha^−1^, to HTP Og 300. However, for all plants treated with the organic fertilizer, the mean weight per fruit was higher, even if not significantly, compared to the plants that received chemical fertilization (Table 1).

The highest yields were obtained for cultivar HTP F_1_, organically fertilized and irrigated according to the 100% evapotranspiration replenishment, followed by Siriana F_1_ cv., chemically fertilized under the same irrigation regime. The lowest yield was recorded for the cultivar IB, chemically fertilized under the 67% water replenishment, which showed a 56.9% decrease compared to the top value. The lowest yield was recorded for the cultivar IB because it is not so well adapted for greenhouses, where the temperatures are higher and the air humidity is lower.

The aforementioned results are consistent with those reported under similar tomato crop management [25,26,27]. 

### 2.2. Fruit Antioxidant Compounds and Activity 

Lycopene is a carotenoid pigment found in red fruits and vegetables, and it is considered a powerful antioxidant [28]. In the present research, its concentration was 50% greater for the cultivars HTP and IB, organically fertilized under 100% evapotranspiration replenishment, compared to the cultivars Siriana F_1_ and HTP, chemically fertilized and watered with 200 m^3^·ha^−1^ (Figure 1).

Regardless of the cultivar, the lowest lycopene values were recorded in the tomato fruits of the plants chemically fertilized and irrigated with 200 m^3^·ha^−1^. The higher lycopene concentration detected in the tomato fruits of the plants treated with organic fertilizer may be due to an intense chemical and biological activity in soils that are organically fertilized compared with soils that are chemically fertilized [21]. However, in other investigations carried out in the Mediterranean area, the lycopene concentration was higher in conventionally grown cherry tomato fruits [9] and under water deficit, depending on the cultivar [24], suggesting that the environment, farming management, and genotype may affect this antioxidant compound. It was also observed that at 472 nm wavelength, the lycopene values for most of the interactions were greater, though with no statistical significance as compared to those obtained at 502 nm. Similar values of the lycopene content were obtained by Inculet et al. [29], and amounts below 6.82 mg·100 g^−1^ f.w. were recorded by Jędrszczyk and Ambroszczyk [30] with the organically fertilized tomato cultivar Mieszko F_1_. As previously reported, the lycopene concentration of tomato fruits can vary with the cultivar, the environmental factors, and the soil’s biological status [31,32,33]. In the present experiment, the fertilization type and the irrigation regime had a significant influence on the lycopene concentration of tomato fruits (Table 3). Regardless of the wavelength used for the determination, the lycopene concentration of the tomato fruits that were organically fertilized was significantly higher as compared with the tomato fruits that were chemically fertilized. Regarding the irrigation regime, significant differences were observed only for the lycopene concentration measured at 502 nm. In this case, the lycopene concentration recorded in the tomato fruits watered with 300 m^3^·ha^−1^ was significantly higher as compared with 200 m^3^·ha^−1^ water. No significant differences were recorded between the cultivars. 

The polyphenol concentration was highest in the cultivar IB, which did not significantly differ from HTP F1, whereas S and IB showed the highest antioxidant activity. The latter was higher under organic management compared to the conventional one. In a previous investigation [34], the organic crop system elicited significantly higher contents of apigenin acetylhexoside, caffeic acid hexoside I, and phloretin dihexoside out of the 30 polyphenols detected in tomato fruits, though the content of polyphenols was also dependent on the cultivar. 

No significant differences were recorded for the rest of the treatments. Regarding the main effects of cultivar, fertilization type, and irrigation regime on the polyphenol concentration, the cultivar had a statistically significant main effect on the synthesis of polyphenols (Table 3). Their concentration in the tomato fruits of the IB cultivar was significantly higher as compared to S or H cultivars. No significant differences were recorded for the polyphenols between the two fertilization types or irrigation regimes (Table 3). The antioxidant activity was significantly influenced by both cultivar and fertilization type (Table 3).

The results regarding the influence of the three experimental factors applied on the polyphenols and antioxidant capacity of tomato fruits are presented in Table 4.

The total polyphenol concentration was significantly higher (31.3%) in the fruits of organically fertilized Siriana F_1_ watered with 200 m^3^·ha^−1^ compared to the same organically fertilized cultivar irrigated with 300 m^3^·ha^−1^. However, the S Og 300 treatment led to a lower polyphenol level than the fruits of the organically fertilized cultivar IB irrigated with the same amount of water (Table 4). 

The best antioxidant activity was recorded for IB and S cultivars as compared with the H cultivar. However, the results regarding the interactions between cultivar, fertilization type, and irrigation regime showed significant differences only for the IB cultivar that was chemically fertilized with a 100% evapotranspiration replenishment or organically fertilized with a 67% evapotranspiration replenishment as compared to the cultivar HTP that was chemically fertilized and watered according to 67% replenishment. Similar values of the antioxidant activity were obtained in previous studies on tomato [6,29] and in related species such as red pepper [35]. Regarding fertilization type, organic fertilization gave a significantly higher antioxidant activity compared to chemical fertilization (Table 3).

### 2.3. Fruit Elemental Composition

The main effects of the experimental factors on the macroelement concentration in tomato fruits are presented in Table 5. The results showed that Mg and K were significantly influenced by the fertilization type only. For both macroelements, Ch fertilization enhanced their concentration in the tomato fruits as compared to Og fertilization. The highest P concentration was affected by both cultivar and fertilization type. For S and H cultivars, the P concentration was significantly higher as compared with the IB cultivar. Regarding fertilization type, the Ch treatment showed a P concentration in the fruits that was statistically higher than Og fertilization. Ca was significantly affected by all the factors applied (cultivar, fertilization, irrigation regime). The cultivar Siriana F_1_ showed a higher Ca value than IB and HTP F_1_. As in the case of Mg, P, and K, the Ch fertilization enhanced the Ca concentration in the tomato fruits, significant differences being recorded as compared to Og fertilization. Additionally, it was observed that with 67% evapotranspiration replenishment (200 m^3^·ha^−1^), the Ca concentration was significantly higher as compared with 100% evapotranspiration replenishment (300 m^3^·ha^−1^).

The results on the interaction between the cultivar, fertilization type, and irrigation regime on the macroelement concentration (magnesium—Mg, phosphorus—P, potassium—K, calcium—Ca) in tomato fruits are presented in Table 6.

The Mg concentration in tomato fruits showed a significant increase of 74.7% over cultivar HTP Og 300 to S Ch 200. 

Chemical fertilization usually resulted in a higher P concentration in tomato fruits compared to organic nutrition, suggesting an easier plant uptake of this element from chemical fertilizers in greenhouses. In particular, the P concentration was 80% higher in the cultivar HTP F_1_ x organic fertilization x 67% ET replenishment compared to the cultivar IB under the same fertilization type and irrigation regime. 

The element found at the highest concentrations in the tomato fruits was K. The concentration ranged from 188.00 mg·100 g^−1^ f.w. for IB Og 300 to 248.67 mg·100 g^−1^ f.w. for IB Ch 200. K was not significantly affected by the interaction between the three experimental factors (Table 6). However, the highest concentration of K was registered for Ch fertilization (Table 5). In previous research [26], the higher effectiveness of chemical fertilization and moderate irrigation regime on potassium content was also reported.

Calcium is a major element in plants, but it is also very important for human health due to its significant concentration in the bones [6,15,16]. In our study, the Ca concentration was 2–3 times lower in the fruits of cultivar HTP F_1_ under organic fertilization with 200 m^3^·ha^−1^ and 300 m^3^·ha^−1^ irrigation regimes than the Ch treatment.

Likewise, chemical fertilization also elicited a higher accumulation of this element in the tomato fruits of cultivar Siriana F_1_ (Table 5). Significant differences were also seen between cultivars Siriana F_1_ and HTP, chemically fertilized, and cultivar IB, chemically or organic fertilized, regardless of the irrigation regime. 

Notably, the mineral concentration recorded in the present study is much higher compared to previous investigations carried out on tomato [27,36]. Indeed, the mineral concentration is closely dependent on the crop system, i.e., the soil and environmental conditions, the genotype, and the farming practices. 

The main effects of the experimental factors on the microelement concentration in tomato fruits are presented in Table 7. In this respect, a significant influence of the cultivar was recorded for Mn, Zn, Ni, and Pb: Mn showed the lowest concentration in Siriana F_1_, whereas the other two cultivars did not differ from each other; the opposite trend was recorded for Pb. For Zn, the lowest concentration was recorded for IB and the highest for Siriana F_1_ and HTP F_1_ cultivars, while for Ni, the lowest concentration was found in the tomato fruits of HTP F_1_ and the highest in Siriana F_1_ and Inima de Bou.

Significant influence of the fertilization type on microelements and heavy metal concentrations were found for Cu, Mn, Ni, and Pb. For all these elements, the lowest values were recorded for Og fertilization as compared to Ch fertilization. No significant influence of the fertilization type was observed for Fe, Zn, and Cr (Table 7).

The irrigation regime had a significant influence on Pb only, the 100% evapotranspiration replenishment (300 m^3^·ha^−1^) giving the lowest concentration of this element in the tomato fruits.

In summary, Cu was influenced by the fertilization type only, Mn and Ni by cultivar and irrigation regime, Pb by all the factors taken into consideration, while Fe and Cr by none of them.

With regard to microelements (Table 8), the Cu concentration was not significantly affected by the interaction between the three experimental factors applied. However, the values were below the tolerance threshold relevant to this element. Slightly higher concentrations of Cu were detected in the tomato fruits obtained in the present experiment compared to previous investigations [35,37], probably due to the organic cultivation technology consisting of antifungal treatments based on Cu.

No significant effect of the interaction between the three experimental factors applied was recorded on the Fe concentration in tomato fruits.

Mn was the least represented element in the tomato fruits, with the lowest concentration in cultivar Siriana F_1_, supplied with organic fertilizer and irrigated with 300 m^3^·ha^−1^, and a 2.94-fold higher level in HTP F_1_ that was chemically fertilized under the 200 m^3^·ha^−1^ irrigation regime.

The Zn concentration in the tomato fruits varied between 169 ppm for IB Og 300 interaction and 299 ppm for S Ch 200. Increased amounts of Zn were recorded for chemical fertilization, regardless of the cultivar or the irrigation regime, with slightly higher but not significant amounts for the irrigation with 200 m^3^·ha^−1^water. For IB Og 300, the Zn values registered were significantly lower compared with S Ch 200, but higher than those reported by Hura et al. [37] or Munteanu et al. [38].

With regard to mineral elements, Ordóñez-Santos et al. [39] reported that the micronutrients analyzed (Fe, Zn, Mn, and Cu) were affected by the genotype, whereas no significant differences were recorded between organic and conventional crop systems, taking into account the efficient soil fertility management. However, Mn was significantly affected by the interaction between the crop system and the cultivar. In other research [40], the cropping method showed a significant interaction with the cultivar on macro- and microelements. In particular, the microelements were more affected by cultivar and the macroelements the cropping method, and only K and Mg gave a remarkable contribution to the mineral intake from tomato fruits.

Among the heavy metals, the Cr concentration of the tomato fruits was not significantly affected by the interaction between the three experimental factors applied. Compared to other tomato cultivars, the Cr concentration recorded in the fruits of the varieties examined in the present trial was higher, especially for the chemically fertilized plants [21].

The Ni and Pb concentrations in the tomato fruits of Siriana F_1_ that was chemically fertilized and irrigated according to the 67% evapotranspiration replenishment were 2.37-fold and 2.85-fold higher, respectively, compared to those measured in HTP F_1_ under organic fertilization and 100% replenishment. In cultivar Siriana F_1_, chemical fertilization elicited a higher fruit Ni and Pb accumulation compared to both organic nutrition and the same nutrition type applied to these other two cultivars, suggesting an important genotype role in the absorption of this element. Lower Pb concentrations in tomato fruits obtained in greenhouses were reported by Munteanu et al. [38].

However, in all the experimental treatments, the concentration of each heavy metal in tomato fruit did not overcome the related acceptance threshold: Cr = 150; Ni = 100; Pb = 50 [38].

Similar results regarding the microelement concentrations in the tomato fruits were obtained by Wang et al. [25] by using a moderate water regime in protected environment conditions.

The higher fruit concentrations of some heavy metals elicited by chemical fertilization, compared to organic fertilization, show that the former type favors a higher plant uptake of potentially toxic elements. Therefore, the chemical fertilizer application should be accurately monitored in order to prevent the excessive accumulation of heavy metals in tomato fruits. 

## 3. Materials and Methods

### 3.1. Plant Material and Growth Conditions

The experiment was done at the University of Agricultural Sciences and Veterinary Medicine (UASMV) of Iasi, Romania (47°19′25″ N, 27°54′99″ E, 150 m a.s.l), at the experimental station in a 400 m^2^ tunnel. The study was conducted over two years, 2018 and 2019, starting from mid-April each year. During the experiment period, the mean temperature was 18.4 °C in 2018 and 18.3 °C in 2019. The sunlight was 244 h in 2018 and 213 h in 2019, with a relative humidity that varied between 67% to 70%. Tomato crop was preceded by runner beans on the soil used in the present investigation. The soil was alluvial cambic chernozem soil with the following characteristics: pH 7.2; 3.2% organic matter; 31% clay; 28 mg·kg^−1^ N, 1.83% K_2_O; 1.13% CaO; 0.19% P_2_O_5_; 0.16% MgO; 0.46% Na_2_O; 3.96% Fe_2_O_3_; 0.11% MnO; 49 ppm Cu; 103 ppm Zn; EC 416 µS·cm^−1^. The soil was not subjected to any fumigation or pesticide treatment.

The experimental protocol was based on the comparison between two irrigation regimes in factorial combination with two fertilization types and three cultivars by using a split-plot design with three replications. The irrigation regime was assigned to the main plot, the fertilization type to the subplot, and the cultivar to the sub-subplot. The experimental unit surface area was 3.6 m^2^ (0.30 m^2^ nutritive surface for each plant), including 12 tomato plants planted on beds with 1 m between rows and 0.3 m between plants per row.

The two irrigation regimes were based on the replenishment of 67% or 100% of ET [41], i.e., on the two watering volumes of 200 and 300 m^3^·ha^−1^ at each irrigation, respectively. Tomato plants were irrigated 26 times over the whole crop season (as an average of the two crop seasons), corresponding to 67% or 100% crop evapotranspiration, respectively.

The two fertilization types consisted of the application of chemical fertilizer (Cristaland, NPK 16-16-16 by MsBiotech, Termoli, Italy) or organic fertilizer (Orgevit^®^ by MeMon BV, Arnhem, The Netherlands). The organic fertilizer is a product based on chicken manure with the following characteristics: pH 7, 4% N, 2.5% P_2_O_5_, 2.3% K_2_O, 1% MgO, 0.02% Fe, 0.01% Mn, 0.01% B, 0.01% Zn, 0.001% Cu, and 0.001% Mo. The fertilizers were applied to the soil at a dose of 800 kg ha^−1^ for the chemical treatment and 4500 kg ha^−1^ for the organic treatment. Both fertilizers were applied three times as follows: 50% of the total amount in coincidence with the final soil preparation prior to planting; 25% when the first fruit of the first cluster reached a 1-cm diameter; the last dose (25%) when the first fruit of the third cluster reached a 1-cm diameter. The doses of fertilizers were calculated by taking into account the following: the chemical composition of each formulate; that 60% of N, P_2_O_5_, and K_2_O contents of the Og (Organic) fertilizer is available for plant assimilation in the year of application; 15% of nitrogen leaching is expected upon the chemical fertilizer supply, as the rate of nutrient release is much lower compared to simple nitrate fertilizers. In particular, the organic fertilizer was supplied three weeks before transplanting in order to allow the onset of the nitrification process, thus preventing the initial ammonium accumulation that may cause toxicity and enhancing the total nitrate amount released to plants over the entire crop cycle ET [42]. 

The three cultivars of *Solanum lycopersicum* were compared: Siriana F_1_ and Inima de Bou, released by the Vegetable Research Station Buzau, and HTP F_1_ by the seed company Hazera. All of them produce round-shaped fruits with a smooth to waved surface.

The following growing practices were applied to all tomato plants: vertical training due to indeterminate growth, pruning above the fifth fruit truss, and treatments against pests with azadirachtin and against diseases with copper-based formulates.

The harvests began on 10 and 13 July and ended on 24 and 26 October in the first and second research years, respectively.

### 3.2. Biometric and Yield Determinations

In each plot and at each harvest, the number and weight of ripe fruits from any truss, as well as the mean fruit weight on random 20-fruit samples, were determined. The yields (kg·ha^−1^) were calculated by using the following formula: (plants/ha × fruits/plant × average fruit weight)/1000 [29]. The plant heights (cm) were measured after the last harvest in each experimental treatment.

### 3.3. Sample Preparation

In each plot, 20-ripe fruit random samples, each of them weighing 3000–3500 g, were collected from the different clusters for laboratory analyses.

The samples needed for biochemical analyses (lycopene and polyphenol contents, antioxidant capacity) were prepared as follows: fruits were cut into 1 cm fragments and dried on a Sanyo stove, type MOV-112F, at a temperature of 70 °C until constant weight. The samples were then ground into small fragments of 0.1–1 mm.

### 3.4. Lycopene Content

The lycopene concentration of the tomato fruits was determined by using the solvent extraction method described by Butnariu and Giuchici [43]. Briefly, 0.6 g of fresh tomato puree was added in 5 mL of 0.05% (*w/v*) butylated hydroxytoluene (BHT) in acetone, 5 ml of 95% ethanol, and 10 ml of hexane and stirred for 15 min on ice. After stirring, 3 ml of deionized water was added, and the samples were stirred for 5 more minutes on ice. For the phase separation, the samples were left at room temperature for 5 min. The absorbance of the upper layer of hexane was measured at 472 and 502 nm by using hexane as a blank.

### 3.5. Polyphenol Content

The concentration of different phenolic compounds in 70% ethanolic extracts was measured using an HPLC–MS method that allows the simultaneous detection of several phenolic compounds with a single column pass through [44]. The detection of the compounds was performed in both UV and MS modes [45]. The UV detector was set at 330 nm for 17.5 min, and then at 370 nm. The mobile phase was a binary gradient, with methanol and acetic acid 0.1% (*v/v*); the flow rate was 1 mL·min^−1^ and the injection volume was 5 µL [46].

### 3.6. Antioxidant Activity

The neutralizing effect of the DPPH free radical was calculated at three different concentrations of methanol and ethanol extracts: 0.05 ml of 10, 5, and 2.5 mg·mL^−1^ extracts were mixed with 2.95 mL solution DPPH. After a 5-min reaction time, the absorption was measured at 420 nm using methanol as a blank. Approximately 10 g of finely ground plant material was extracted in 100 mL of 80% aqueous solution of ethanol at room temperature for 1 h. The extracts were filtered, and the filters were left to evaporate in a dry environment. The percentage of free-radical scavenging activity was determined using the following formula: 100 × (Ai–Af)/Ai, where Ai = the absorption before the addition of the tested extract; Af = the absorption value after 5 min reaction time. 

QE was used as a positive control. At a concentration of 2.5 mM, QE was able to fully neutralize the level of DPPH radical used [47]. 

### 3.7. Mineral Content

The mineral concentration (macro- and microelements) of the tomato fruits was determined by using the atomic absorption spectrometry method. The fruits were first oven-dried at 105 °C for 48 h, and then samples of 0.5 and 1 g were digested [9]. The solutions obtained were analyzed by the atomic-absorption spectrometer Contra 300 (Analytik Jena, Göettingen, Germany).

### 3.8. Statistical Analysis of the Data

The experimental results are expressed as means ± SD. The data were statistically processed by three-way ANOVA upon verifying the normality through the Shapiro–Wilk test, and Tukey’s test was performed for mean separation at *p* ≤ 0.05, using SPSS software version 21 (IBM Microsoft, New York, USA). 

## 4. Conclusions

Tomato yield and the number of fruits were significantly influenced by cultivar but not by fertilization treatments. The treatment with a greater supply of water led to higher fruit yield. Indeed, the highest yield was achieved in the combination of the HTP F_1_ cultivar under organic fertilization and 100% evapotranspiration replenishment.

The concentrations of phosphorus, calcium, nickel, and lead in the fruits of the Siriana F_1_ cultivar were higher under the 67% evapotranspiration replenishment and chemical fertilization, which suggest that these mineral elements are more easily absorbed by this cultivar as compared with HTP F_1_ and Inima de Bou cultivars.

Macroelement concentrations of manganese and potassium in tomato fruits were affected by the fertilization type, phosphorus by cultivar and fertilization type, and calcium by all the factors examined. Microelement concentrations in tomato fruits were influenced by the treatments. Copper was affected by the fertilization type only, manganese and nickel by the cultivar and irrigation regime, and lead by all three factors. The positive effect of organic fertilization on the lycopene accumulation and antioxidant activity of tomato fruits showed that this fertilization type can be used as an alternative to chemical fertilization in sustainable agriculture, giving premium quality products.

## Figures and Tables

**Figure 1 plants-09-01053-f001:**
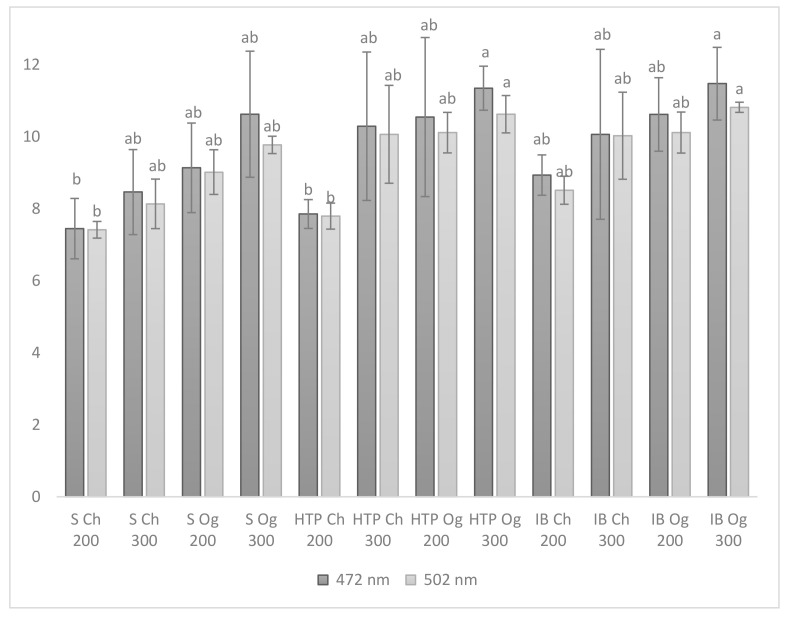
Influence of cultivar, fertilization type, and irrigation regime on the lycopene concentration at 472 and 502 nm (mg·100 g^−1^f.w.). S—Siriana F_1_; H—HTP F_1_; IB—Inima de Bou; Ch—chemical; Og—organic; 200—200 m^3^ water·ha^−1^; 300—300 m^3^ water·ha^−1^. Different letters represent significant differences between the treatments, according to Tukey’s test at *p* ≤ 0.05.

**Table 1 plants-09-01053-t001:** Influence of the experimental treatments on plant height and yield components.

Treatment	Plant Height at Last Harvest (cm)	Number of Fruits per Plant	Mean Weight Per Fruit (g)	Yield (t·ha^−1^)
Year				
2018	217.0	23.7	167.6	135.6
2019	213.8	23.3	167.0	134.0
	ns	ns	ns	ns
Cultivar				
Siriana F_1_ (S)	206.4 ± 14.6	24.1 ± 1.64 a	161 ± 11	134.4 ± 7.0 ab
HTP F_1_ (H)	215.6 ± 15.4	24.3 ± 1.65 a	183 ± 13	152.4 ± 8.4 a
Inima de Bou (IB)	224.2 ± 15.9	22.0 ± 1.50 b	158 ± 11	117.7 ± 9.1 b
	ns		ns	
Fertilization				
Chemical (Ch)	227.0 ± 16.1	24.2 ± 1.7	160 ± 11	130.9 ± 10.2
Organic (Og)	204.4 ± 14.5	22.7 ± 1.6	175 ± 12	134.3 ± 10.4
	ns	ns	ns	ns
Irrigation regime				
200 m^3^·ha^−1^	215.6 ± 15.3	22.4 ± 1.5 b	160 ± 11	121.0 ± 9.4 b
300 m^3^·ha^−1^	215.8 ± 15.3	24.5 ± 1.7 a	175 ± 12	144.3 ± 11.2 a
	ns		ns	

200 m^3^·ha^−1^ (67% evapotranspiration (ET) replenishment); 300 m^3^·ha^−1^ (100% ET replenishment); ns: not statistically significant. Within each column and each experimental factor, different letters mean significant differences between the treatments, according to Tukey’s test at *p* ≤ 0.05.

**Table 2 plants-09-01053-t002:** Interaction between cultivar, fertilization type, and irrigation regime on plant height and yield components.

Treatment	Plant Height at Last Harvest (cm)	Number of Fruits Per Plant	Mean Weight Per Fruit (g)	Yield (t·ha^−1^)
S Ch 200	216.83 ± 15.38	23.56 ± 1.60 ab	148 ± 1.00 b	117.51 ± 9.13 ab
S Ch300	220.90 ± 15.67	28.41 ± 1.93 a	160 ± 1.10 ab	153.19 ± 11.90 ab
S Og200	199.52 ± 14.16	19.41 ± 1.32 b	163 ± 1.10 ab	106.61 ± 8.28 ab
S Og300	188.32 ± 13.36	25.04 ± 1.70 ab	174 ± 1.21 ab	146.89 ± 11.41 ab
HTP Ch200	236.17 ± 16.76	24.97 ± 1.70 ab	168 ± 1.15 ab	141.37 ± 10.98 ab
HTP Ch300	221.92 ± 15.74	25.19 ± 1.71 ab	171 ± 1.15 ab	145.16 ± 11.27 ab
HTP Og200	204.61 ± 14.52	24.63 ± 1.68 ab	184 ± 1.25 ab	152.72 ± 11.86 ab
HTP Og300	203.59 ± 14.44	22.26 ± 1.52 ab	209 ± 1.43 a	157.12 ± 12.20 a
IB Ch200	235.15 ± 16.68	20.14 ± 1.37 b	147 ± 1.01 b	100.11 ± 7.77 b
IBCh300	231.08 ± 16.39	22.78 ± 1.55 ab	167 ± 1.15 ab	128.20 ± 9.96 ab
IB Og 200	201.56 ± 14.30	21.56 ± 1.47 ab	148 ± 1.00 b	107.53 ± 8.35 ab
IB Og 300	229.04 ± 16.25	23.44 ± 1.60 ab	171 ± 1.15 ab	135.08 ± 10.49 ab
	ns			

S—Siriana F_1_; H—HTP F_1_; IB—Inima de Bou; Ch—chemical; Og—organic; 200—200 m^3^·ha^−1^ (67% ET replenishment); 300—300 m^3^·ha^−1^ (100% ET replenishment); ns: not statistically significant. Within each column, different letters mean significant differences between the treatments, according to Tukey’s test at *p* ≤ 0.05.

**Table 3 plants-09-01053-t003:** Influence of the cultivar, fertilization type, and irrigation regime on the lycopene, polyphenols, and antioxidant activity.

Treatment	Lycopene (mg·100 g^−1^ f.w.)		Polyphenols (mg·100 g^−1^ f.w.)	Antioxidant Activity (mmol Trol·100 g^−1^ f.w.)
	472 nm	502 nm		
Cultivar				
Siriana F_1_ (S)	8.91 ± 0.55	8.58 ± 0.14	1584.95 ± 70.11 b	78.09 ± 3.91 a
HTP F_1_ (H)	10.01 ± 0.15	9.70 ± 0.64	1706.53 ± 1.52 ab	57.97 ± 0.31 b
Inima de Bou (IB)	10.27 ± 0.70	9.85 ± 0.14	1801.71 ± 34.83 a	78.80 ± 0.99 a
	ns	ns		
Fertilization				
Chemical (Ch)	8.84 ± 0.61 b	8.66 ± 0.24 b	1702.24 ± 61.83	68.38 ± 1.80 b
Organic (Og)	10.62 ± 0.21 a	10.10 ± 0.21 a	1693.22 ± 20.27	74.87 ± 1.36 a
			ns	
Irrigation regime				
200 m^3^·ha^−1^	9.09 ± 0.20	8.83 ± 0.12 b	1736.72 ± 44.28	70.98 ± 1.76
300 m^3^·ha^−1^	10.38 ± 0.74	9.93 ± 0.22 a	1658.74 ± 62.28	72.26 ± 2.80
	ns		ns	ns

S—Siriana F_1_; H—HTP F_1_; IB—Inima de Bou; Ch—chemical; Og—organic; 200—200 m^3^·ha^−1^ (67% ET replenishment); 300—300 m^3^·ha^−1^ (100% ET replenishment); ns: not statistically significant. Within each column and within each experimental factor, different letters mean treatments differ significantly according to Tukey’s test at *p* ≤ 0.05

**Table 4 plants-09-01053-t004:** Interaction between cultivar, fertilization type, and irrigation regime on total polyphenol concentration and antioxidant capacity.

Treatment	Polyphenols (mg·100 g^−1^ f.w.)	Antioxidant Activity (mmol Trol·100 g^−1^ f.w.)
S Ch 200	1623.26 ± 135.06 ab	77.94 ± 6.48 ab
S Ch 300	1724.12 ± 151.78 ab	71.98 ± 6.34 ab
S Og 200	1869.19 ± 146.27 a	82.81 ± 6.48 ab
S Og 300	1423.25 ± 152.14 b	79.63 ± 10.79 ab
HTP Ch 200	1609.27 ± 89.91 ab	52.14 ± 2.91 b
HTP Ch 300	1686.29 ± 142.13 ab	54.84 ± 4.62 ab
HTP Og 200	1724.27 ± 118.01 ab	61.45 ± 4.21 ab
HTP Og 300	1806.28 ± 88.81 ab	63.46 ± 3.12 ab
IB Ch 200	1826.26 ± 211.86 ab	65.24 ± 7.57 ab
IB Ch 300	1744.23 ± 211.59 ab	88.11 ± 10.69 a
IB Og 200	1768.08 ± 100.29 ab	86.31 ± 4.90 a
IB Og 300	1868.29 ± 110.09 a	75.54 ± 4.45 ab

S—Siriana F_1_; H—HTP F_1_; IB—Inima de Bou; Ch—chemical; Og—organic; 200—200 m^3^·ha^−1^ (67% ET replenishment); 300—300 m^3^·ha^−1^ (100% ET replenishment). Within each column, different letters mean significant differences between the treatments, according to Tukey’s test at *p* ≤ 0.05.

**Table 5 plants-09-01053-t005:** Influence of cultivar, fertilization type, and irrigation regime on macroelements concentration in tomato fruits.

Treatment	Mg (mg·100 g^−1^ f.w.)	P (mg·100 g^−1^ f.w.)	K (mg·100 g^−1^ f.w.)	Ca (mg·100 g^−1^ f.w.)
Cultivar				
Siriana F_1_ (S)	11.84 ± 0.79	10.82 ± 0.26 a	227.42 ± 15.03	12.32 ± 0.29 a
HTP F_1_ (H)	9.76 ± 0.25	12.44 ± 0.22 a	217.41 ± 4.11	10.41 ± 0.28 b
Inima de Bou (IB)	10.30 ± 0.48	8.51 ± 0.57 b	209.50 ± 2.13	5.88 ± 0.39 c
	ns		ns	
Fertilization				
Chemical (Ch)	11.71 ± 0.22 a	13.03 ± 0.57 a	230.61 ± 7.46 a	11.99 ± 0.05 a
Organic (Og)	9.55 ± 0.51 b	8.14 ± 0.15 b	205.61 ± 4.71 b	7.08 ± 0.30 b
Irrigation regime				
200 m^3^·ha^−1^	11.07 ± 0.19	11.36 ± 0.60	225.61 ± 9.10	10.07 ± 0.13 a
300 m^3^·ha^−1^	10.19 ± 0.53	9.81 ± 0.32	210.61 ± 1.49	9.00 ± 0.33 b
	ns	ns	ns	

S—Siriana F_1_; H—HTP F_1_; IB—Inima de Bou; Ch—chemical; Og—organic; 200—200 m^3^·ha^−1^ (67% ET replenishment); 300—300 m^3^·ha^−1^ (100% ET replenishment); ns: not statistically significant. Within each column and experimental factor, different letters mean significant differences between the treatments, according to Tukey’s test at *p* ≤ 0.05.

**Table 6 plants-09-01053-t006:** Interaction between cultivar, fertilization type, and irrigation regime on macroelements concentration in tomato fruits.

Treatment	Mg (mg·100 g^−1^ f.w.)	P (mg·100 g^−1^ f.w.)	K (mg·100 g^−1^ f.w.)	Ca (mg·100 g^−1^ f.w.)
S Ch 200	13.10 ± 1.04 a	14.33 ± 0.79 ab	247.33 ± 21.67	15.93 ± 1.34 a
S Ch 300	12.10 ± 1.66 ab	13.17 ± 1.12 ac	238.33 ± 18.55	14.20 ± 0.96 ab
S Og 200	11.60 ± 0.64 ab	8.30 ± 0.59 ce	219.00 ± 29.61	9.63 ± 0.46 bc
S Og 300	10.50 ± 0.90 ab	7.47 ± 0.38 de	205.00 ± 11.68	9.50 ± 1.10c
HTP Ch 200	11.40 ± 0.79 ab	15.67 ± 1.82 a	227.00 ± 19.08	15.00 ± 1.82 a
HTP Ch 300	10.60 ± 0.52 ab	13.90 ± 1.70 ab	217.00 ± 14.93	14.20 ± 0.79 ab
HTP Og 200	9.50 ± 1.10 ab	11.70 ± 0.64 ad	215.67 ± 18.22	7.83 ± 0.65 cd
HTP Og 300	7.50 ± 0.89 b	8.47 ± 0.49 cde	210.00 ± 14.36	4.60 ± 0.40 d
IB Ch 200	11.60 ± 0.95 ab	11.63 ± 1.02 ad	248.67 ± 12.14	6.50 ± 0.49 cd
IB Ch 300	11.40 ± 1.00 ab	9.50 ± 0.72 be	205.33 ± 23.84	6.10 ± 0.80 cd
IB Og 200	9.20 ± 0.72 ab	6.50 ± 0.90 e	196.00 ± 23.69	5.50 ± 0.31 cd
IB Og 300	9.00 ± 1.21 ab	6.40 ± 0.36 e	188.00 ± 10.79	5.40 ± 0.45 cd
			n.s.	

S—Siriana F_1_; H—HTP F_1_; IB—Inima de Bou; Ch—chemical; Og—organic; 200—200 m^3^·ha^−1^ (67% ET replenishment); 300—300 m^3^·ha^−1^ (100% ET replenishment); ns: not statistically significant. Within each column, different letters mean significant differences between the treatments, according to Tukey’s test at *p* ≤ 0.05.

**Table 7 plants-09-01053-t007:** Influence of cultivar, fertilization type, and irrigation regime on microelement and heavy metal concentrations in tomato fruits.

Treatment	Cu (mg·kg^−1^ f.w.)	Fe (mg·kg^−1^ f.w.)	Mn (mg·kg^−1^ f.w.)	Zn (mg·kg^−1^ f.w.)	Cr (mg·kg^−1^ f.w.)	Ni (mg·kg^−1^ f.w.)	Pb (mg·kg^−1^ f.w.)
Cultivar							
Siriana F_1_ (S)	40.00 ± 1.84	57.74 ± 0.85	2.77 ± 0.10 b	252.50 ± 16.49 a	100.75 ± 0.80	65.00 ± 1.31 a	62.00 ± 4.18 a
HTP F_1_ (H)	36.50 ± 0.44	50.94 ± 2.71	4.82 ± 0.06 a	223.00 ± 2.61 a	91.00 ± 1.21	40.00 ± 0.89 b	35.25 ± 0.97 b
Inima de Bou (IB)	39.50 ± 0.75	50.53 ± 2.83	4.65 ± 0.09 a	175.50 ± 3.96 b	95.75 ± 10.98	56.00 ± 3.77 a	45.00 ± 0.83 b
	ns	ns			ns		
Fertilization type							
Chemical (Ch)	42.33 ± 0.50 a	55.82 ± 3.22	4.15 ± 0.16 a	232.00 ± 2.79	101.67 ± 3.53	59.34 ± 3.16 a	54.00 ± 1.06 a
Organic (Og)	35.00 ± 0.80 b	50.30 ± 1.07	3.67 ± 0.01 b	202.00 ± 13.31	90.00 ± 5.20	48.00 ± 0.76 b	40.83 ± 2.01 b
		ns		ns	ns		
Irrigation regime							
200 m^3^·ha^−1^	39.34 ± 1.22	54.05 ± 3.34	4.00 ± 0.14	225.83 ± 5.39	98.00 ± 1.14	56.34 ± 1.64	50.67 ± 1.35 a
300 m^3^·ha^−1^	38.00 ± 1.53	52.07 ± 1.10	3.82 ± 0.13	208.17 ± 10.38	93.66 ± 7.13	51.00 ± 2.30	44.16 ± 1.62 b
	ns	ns	ns	ns	ns	ns	

S—Siriana F_1_; H—HTP F_1_; IB—Inima de Bou; Ch—chemical; Og—organic; 200—200 m^3^·ha^−1^ (67% ET replenishment); 300—300 m^3^·ha^−1^ (100% ET replenishment); ns: not statistically significant. Within each column and within a treatment, different letters mean treatments differ significantly according to Tukey’s test at *p* ≤ 0.05.

**Table 8 plants-09-01053-t008:** Interaction between cultivar, fertilization type, and irrigation regime on microelement and heavy metal concentrations in tomato fruits.

Treatment	Cu (mg·kg^−1^ f.w.)	Fe (mg·kg^−1^ f.w.)	Mn (mg·kg^−1^ f.w.)	Zn (mg·kg^−1^ f.w.)	Cr (mg·kg^−1^ f.w.)	Ni (mg·kg^−1^ f.w.)	Pb (mg·kg^−1^ f.w.)
S Ch 200	47 ± 3.91	65.1 ± 3.7	2.9 ± 0.26 bc	299 ± 23.39 a	105 ± 8.2	90 ± 5.03 a	77 ± 6.03 a
S Ch 300	46 ± 4.05	61.9 ± 5.2	2.6 ± 0.23 c	253 ± 34.27 ab	103 ± 14.0	67 ± 5.65 ab	75 ± 10.16 ab
S Og 200	35 ± 2.74	52.4 ± 3.6	1.9 ± 0.15 c	232 ± 12.96 ab	99 ± 5.5	53 ± 3.63 bc	52 ± 2.90 bc
S Og 300	32 ± 4.33	51.5 ± 2.5	1.7 ± 0.25 c	226 ± 19.05 ab	96 ± 8.1	50 ± 2.46 bc	44 ± 3.71 cd
HTP Ch 200	39 ± 2.18	53.1 ± 6.2	5.0 ± 0.31 a	253 ± 17.31 ab	98 ± 6.7	42 ± 4.87 c	43 ± 2.94 cd
HTP Ch 300	38 ± 3.20	51.7 ± 6.3	4.9 ± 0.42 a	226 ± 11.11 ab	92 ± 4.5	41 ± 4.97 c	39 ± 1.91 cd
HTP Og 200	35 ± 2.40	50.9 ± 4.5	4.8 ± 0.32 a	218 ± 25.29 ab	88 ± 7.3	39 ± 3.24 c	32 ± 3.71 cd
HTP Og 300	34 ± 1.67	48.1 ± 3.8	4.6 ± 0.23 ab	195 ± 10.89 b	86 ± 7.6	38 ± 3.34 c	27 ± 3.28 d
IB Ch 200	43 ± 4.99	52.4 ± 7.1	4.9 ± 0.57 a	181 ± 15.26 b	112 ± 8.8	59 ± 4.62 bc	50 ± 2.84 cd
IB Ch 300	41 ± 4.97	50.8 ± 2.8	4.7 ± 0.58 a	180 ± 12.32 b	100 ± 13.5	57 ± 7.72 bc	40 ± 3.33 cd
IB Og 200	37 ± 2.09	50.4 ± 4.2	4.5 ± 0.26 ab	172 ± 8.46 b	86 ± 4.8	55 ± 3.07 bc	50 ± 4.40 cd
IB Og 300	37 ± 2.18	48.5 ± 3.3	4.5 ± 0.25 ab	169 ± 19.61 b	85 ± 7.2	53 ± 4.47 c	40 ± 3.13 cd
	n.s.	n.s.			n.s.		

Cv—cultivar; S—Siriana F_1_; H—HTP F_1_; IB—Inima de Bou; Ch—chemical; Og—organic; 200—200 m^3^·ha^−1^ (67% ET replenishment); 300—300 m^3^·ha^−1^ (100% ET replenishment); ns: not statistically significant. Within each column, different letters mean significant differences between the treatments, according to Tukey’s test at *p* ≤ 0.05.

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
