# Peer review of "Yield and Nutritional Response of Greenhouse Grown Tomato Cultivars to Sustainable Fertilization and Irrigation Management"

_plants, 2020, doi:10.3390/plants9081053_

Round 1
Reviewer 1 Report
Report on Plants 872400
I am sorry, but this is still not in a form suitable for publication. See my report on the previous version, and my follow-up notes in CAPITALS
Report on v2 Plants 821179
This is an improved version of the manuscript, but still not publishable. THE SAME
The authors have not paid attention to all of my requested changes, and have not indicated why [for example, ‘weight per fruit’ is not used, nor is ‘concentration’, and they must be]. MANY INSTANCES OF ‘CONTENT’ STILL REMAIN, AND THESE MUST BE CHANGED TO READ ‘CONCENTRATION’.
Again, I have annotated the text.
Major issues relate to the convoluted expression in English and sentences or phrases with no sense [and it is not my job to rewrite the text] STILL SOME PARTS ARE POORLY PRESENTED, to the lack of data [a major omission] on main effects of treatments on lycopene, polyphenol content and antioxidant activity and on macro and micro-nutrient parameters, TO BE COMPLETE, THE AUTHORS MUST COMMENT ON THE MAIN EFFECTS OF EACH OF THE THREE TREATMENTS ON LYCOPENE, POLYPHENOL, MACRO AND MICRO NUTRIENTS – IF THEY DON’T THEN THIS PAPER MUST NOT BE PUBLISHED, and a few other issues, such as the very low SEs in Table 1 for mean weight per fruit but lack of treatment effects HOW CAN THE DIFFERENCE BETWEEN TREATMETNS NOT BE SIGNIFICANT IF THE DIFFERENCE IS ABOUT 20 TIMES THE SIZE OF THE STANDARD ERROR, some lack of consistency between the text and data presented [e.g., lines 132-134 and Table 1; lines 174-175 and Table 3..], SEE THE ABSTRACT THAT SAYS NO EFFECT OF IRRIGATION ON YIELD THEN TABLE 1, AND LINES 105 AND 361-362 THAT SAY THERE IS AN EFFECT and really no conclusions in the Conclusions section other than the last sentence REALLY THE CONCLUSIONS ARE ONLY A REPEAT OF THE RESULTS..., AND THEY ARE CONVOLUTED TOO..
I HAVE NOT RE- REVIEWED THE REFERENCES.
BACK TO THE AUTHORS.. THE MANUSCRIPT CANNOT BE PUBLISHED WITH THESE GLARING MISTAKES AND OMISSIONS.

Author Response
Answer to Reviewer 1 (red)
Line 24: “crops” from “tomato crops grown” was deleted as suggested by Reviewer 1.
Line 27: “content” was replaced with “concentration”
Lines 25: “effectiveness” was replaced as suggested with: “effect”.
Line 30: “…did not enhance either the yield or the microelements absorbed by the plants” was rephrased as follows: “did not enhance the concentration of most microelements absorbed by plants. However, the effect of irrigation on yield was significant.”
Line 33: “are easier” was replaced with “are more easily”.
Line 54: “per shoot” was added; “the” from “the red basil” was deleted. “(Ocimum basilicum 'Purpurascens')” was added.
Line 64: “[17,4] was moved to line 63 as suggested.
Lines 93 – 95: As suggested by reviewer 1, the lines from a previous version as corrected by him was added: “As when were no interactions between the year of research and the three experimental factors applied for any measured parameters, all the values are reported as means of the two investigation years.”
Line 97: “contrastingly” was replaced with “contrasting”.
Line 99: “higher” was replaced with “greater”.
Line 116: “is” was replaced with “are”.
Line 130: “..it was observed a tendency for most..” was changed to “a tendency was observed for most”.
Line 131: “to have fewer” was added as suggested. “was lower” was deleted.
Line 136: Lines 135 – 137 (or 131 -133 from the reviewed version) were not commenting Table 1, but Table 2. To be clearer the following changes were made: “it was observed that the mean weight per fruit was higher, even if not in a significant way,..”. Also, “(Table 2)’ was added.
Line 138: “results” was deleted; “yield” was changed to “yields”.
Line 142: “a trait of the population obtained through…” was replaced with “due to its selection, being an open-pollinated”.
Line 176: “content” was replaced with “concentration”
Line 182: “increased” was replaced with “greater by”
Line 189: “content” was replaced with “concentration”
Line 193: “content” was replaced with “concentration”
Line 198: “content” was replaced with “concentration”
Line 201: “content” was replaced with “concentration”
Line 213: “content” was replaced with “concentration”
Line 206: to be clearer “The content ranged from 188.00 mg·100 g-1 f.w. for the interaction between IB Og and 300 m3·ha-1 up to 248.67 mg·100 g-1 f.w. for IB cultivar, Ch fertilized, irrigated with 200 m3·ha-1.”” was rewrite as follows: “The content ranged from 188.00 mg·100 g-1 f.w. for IB Og 300 and 248.67 mg·100 g-1 f.w. for IB Ch 200.”
Line 213: “content” was replaced with “concentration”
Line 218: To be clearer statement: “Significant differences were also seen between S Ch, HTP Ch and IB Ch or Og regardless the irrigation regime.” was rephrased to: “Significant differences were also seen between cultivars Siriana F1 and HTP, chemically fertilized and cultivar IB chemically or organic fertilized, regardless the irrigation regime.”
Line 230: “content” was replaced with “concentration”
Line 236: “content” was replaced with “concentration”
Line 240: “content” was replaced with “concentration”
Line 246: “content” was replaced with “concentration”
Line 248: “content” was replaced with “concentration”
Lines 262 – 264 “Similar results regarding the microelement concentrations were obtained by Wang et al., by using moderate water regime for the tomato plants grown in protected area [6].” Were moved to lines 259-260 and changed as follows: “Similar results regarding the microelement concentrations in the tomato fruits were obtained by Wang et al., by using a moderate water regime in the conditions of protected area [6].”
Line 364: To be clearer “Higher contents of phosphorus and calcium under the chemical fertilization compared to the organic one, and of nichel and lead just in the fruits of cultivar Siriana F1under the 67% evapotranspiration replenishment and chemically fertilized, compared to the organic fertilization, suggest that these mineral elements are easier to be absorbed by plants.” was rephrased as follows “Higher concentrations of phosphorus, calcium, nickel and lead in the fruits of cultivar Siriana F1 under the 67% evapotranspiration replenishment and chemical fertilization, suggest that these mineral elements are easily absorbed by plants.”
Line 367: “at” was replaced with “but under”.
Line 368: the following statement: “indicate that higher irrigation volumes” was deleted.
Line 371: “the” from “the sustainable” was deleted.
- why the differences are not significant given that the errors are small in Table 1.
Dear Editor, we are sorry that the values of standard deviations relevant to the Mean weight per fruit were mistakenly reported in Table 1 and now have been corrected (L 101-102).
In general, the percentage ratios between the standard deviations and the values of the related variables are close to the percentage difference between the treatments within a variable, so that in our opinion it does not sound a surprise that in those cases no significant differences arose.
- why the mineral content is higher than in other studies
L 221-227 Dear Editor, in our opinion the mineral content is closely dependent on the crop system, i.e. the soil and environmental conditions, the genotype and the farming practices. In the latter respect, we managed the irrigation and fertilization in particular ways, different from other trials: a) we did not apply a leaching fraction when irrigating, but we even replenished just the 67% of the evapotranspiration in one of the two irrigation regimes, so that the amount of nutrients available was not lowered; b) we fertilized the plants based on the calculated nutrient requirements, thus providing the actual amount needed.

Reviewer 2 Report
I agree with the corrections in the document, but I found another error.
In the original text in chapter 3.5. you write that the polyphenol content (line 348) was determined according to the authors cited in reference number [56] (Mocan, A .; Vodnar, DC; Vlase, L .; Cris, an, O .; Gheldiu, A.-M .; Cris , and GG) [57] (Vlase, L .; Benedec, D .; Hanganu, D .; Damian, G .; Csillag, I .; Sevastre, B .; Mot, AC; Silaghi - Dumitrescu, 562 R., Tilea, I.) and [58] (Lotfi, S .; Kordsardouei, H .; Oloumi, H.). In the corrected text for the determination of polyphenols (line 326), the collective of Butnaria and Giuchici is mentioned under the serial number [39], but in the references under this number the collective of the authors is Mocan, A .; Vodnar, D.C .; Vlase, L .; Cris, an, O; Gheldiu, A.-M .; Cris, an, G. Specify according to which authors you determined polyphenols!
After the correction, I support the publication of the document.
Author Response
Answer to Reviewer 2
The lycopene content of the tomato fruits was determined under method described by Butnariu and Giuchici at 38 and is correct.
The polyphenols was described by Mocan et al (39), Vlase et al., (40) and Lotfi et al (41)
The references’ numbers were addressed with red color

Reviewer 3 Report
The effect of fertilization and irrigation on certain tomato quality parameters is not a novelty. There is an extensive bibliography on the subject. However, the work reports the results of an experimentation conducted in a specific Romanian area with genotypes adapted to that environment and to the protected conditions adopted. It is therefore a source of further knowledge about tomato cultivation that is worth examining.
The work presents some important flaws which require important revisions.
English language is not fluent and must be improved by a professional editor.
In the abstract there is a description of the work, but the conclusions are lacking
The introduction is a bit confusing and not perfectly focused on the subject. Moreover, the bibliography should be carefully reviewed: many references are not appropriate and in some cases their citation does not correspond to the statement they refer to
Materials and Methods need to be reviewed as they are not always accurate and, in some cases, excessively synthetic.
The presentation of the results should be more organically and better arranged. Often the comments are focuses on specific aspects without identifying the underlying pattern determined by the application of the treatments. Perhaps it would be more useful to treat cv by cv discussing their behavior in response to fertilization and irrigation.
For a more rational analysis I would strongly suggest to consult some works such as:
- Ordóñez‐Santos, L. E., Vázquez‐Odériz, M. L., & Romero‐Rodríguez, M. A. (2011). Micronutrient contents in organic and conventional tomatoes (Solanum lycopersicum L.). International Journal of Food Science & Technology, 46(8), 1561-1568.
- Suárez, M. H., Rodríguez, E. R., & Romero, C. D. (2007). Mineral and trace element concentrations in cultivars of tomatoes. Food Chemistry, 104(2), 489-499.
The discussion is almost non-existent. It is only a description (very general and not much coherent) of the data shown.
Consequently, conclusions should be rewritten after the revision of the manuscript.
The formatting of the references must be checked carefully.
All specific comments are in the attached pdf

Reviewer 4 Report
The manuscript needs minor changes and corrections as outlined below.
Title
Please, modify as follows:
‘Yield and nutritional response of greenhouse grown tomato cultivars to sustainable fertilization and irrigation management’
Abstract
Please, modify as follows (L 21-31):
‘The aim of this study was to compare the effects of two fertilization types (chemical or organic) and two irrigation regimes (67% or 100% of evapotranspiration replenishment) on morphological, biochemical and yield parameters of three indeterminate cultivars of tomato grown in a greenhouse. The results showed that the effect of organic fertilization was not significantly different from that of chemical fertilization, for most of the parameters analyzed. The positive effect of the organic fertilization on the lycopene and polyphenols concentration as well as on antioxidant activity of tomato fruits showed that it can be used as an alternative to chemical fertilization in sustainable horticulture. In each cultivar under the same fertilization type, the effect of irrigation was significant on yield, but the 100% evapotranspiration replenishment did not enhance the fruit concentration of most microelements compared to 67% irrigation regime.’
Key words
Modify as follows:
‘Solanum lycopersicum L., organic fertilization, irrigation regime, production, antioxidants, mineral elements
Introduction
Delete the following sentence (L 40-42):
‘tomato represents the main crop cultivated in greenhouses, thanks to its high economic value based on a higher income and a better-balanced expense compared to other vegetable crops. Annually’
Modify as follows (L 88):
‘in factorial combination with 67% and 100% evapotranspiration replenishment, on yield, growth, antioxidants and elemental composition’
Results and Discussion
In the legend of Table 1, modify as follows:
‘200 m3·ha−1 (67% ET replenishment); 300 m3·ha−1 (100% ET replenishment); ns: not statistically significant; *significant at p≤0.05. Within each column, different letters mean significant differences between the treatments, according to Tukey test at p≤0.05.’
In the legends of Table 2-5, modify as follows:
‘Cv – cultivar; S - Siriana F1; H - HTP F1; IB - Inima de Bou; Ch – chemical; Og - organic; 200 - 200 m3·ha−1 (67% ET replenishment); 300 - 300 m3·ha−1 (100% ET replenishment); ns: not statistically significant; *significant at p≤0.05. Within each column, different letters mean significant differences between the treatments, according to Tukey test at p≤0.05.’
Correct as follows (L 132):
‘to have a lower fruit number’
Delete the following sentence (L 136-138):
‘However, for all plants treated with the organic fertilizer it was observed that the mean weight per fruit was higher, even if not in a significant way, compared to the plants which received the chemical fertilization (Table 2).’
Amend as follows (L 139):
‘The highest yields’
Materials and Methods
Complete the sentence as follows (L 293):
‘evapotranspiration (ET) [37]’
Conclusions
Modify as follows (L 371-372):
‘Cultivar HTP F1 under organic fertilization and 100% evapotranspiration replenishment showed the highest yield.
Round 2
Reviewer 1 Report
Report on Plants 872400 V2
THIS IS GETTING TOO REPETITIVE, AGAIN AN IMPROVED VERSION, BUT STILL LACKING.
FOR EXAMPLE, EVEN THE Abstract writes of ‘content’ and I have indicated that this is the wrong word, and must be replace with ‘concentration’
Likewise, in the Caption for Table 5, and in lines 208-213, so the authors can quite simply do a search and change!!!
They can also provide Tables that indicate the main effects of the treatments on lycopene, polyphenol concentration and antioxidant activity and on macro and micro-nutrient parameters. They did this in Table 1 for main effects on the respective variables, with interactions presented in Table 2 and Figure 1, so they must do likewise for those variables for which they have not presented main effects and write something on the actual effects. Without this I recommend NOT publishing. If the authors can indicate that there were no significant main effects, then fine, but indicate so in the text..
Other changes required:
Lines 26/27 to read ..lycopene and polyphenol concentration..
Lines 28-34 would be more easily verified if main effects are presented.
Line 93 to read As there were no interactions..
Line 131-133 remove the ‘that’ in line 132 and emphasise that this effect is ns (and refer to Table 1)
Lines 136-138 remove ‘it was observed’. Replace ‘in a significant way’ by ‘significantly’, and change reference from Table 2 to Table 1 where this effect is clearly presented!!!!
Line 143 remove ‘an’
Line 225-229. I recommend removing this, for normally fertilization practices are based upon plant requirements, and point a) to me doesn’t make any sense. Indeed it confuses the issue for any reader.
Line 373 Higher than what other treatment? The implication is compared to organic fertilizer with 100% replenishment. Is this what you mean? And is it so important and a good conclusion for only one of the three cultivars tested? Rethink this please. As presented in the previous sentences, this does not suggest that ‘these mineral elements are easily absorbed by plants’.
The whole of the Conclusions needs rethinking and rewriting.
Reviewer 3 Report
After the extensive revision, the manuscript shows an improvement.
There are, however, still some points to be addressed. The additional comments are reported in the attached pdf.
The language still needs an in-depth revision

Author Response
Reviewer 3
Line 23. The comma was deleted
Line 51 compound as deleted
Line 55 was addressed depending on the cultivars used, but in the latest high-lycopene (HLY) lines the level of lycopene can be significantly higher [7].
Was addressed the bibliography Ilahy R., Siddiqui M.W., Tlili I., Montefusco A., Piro G., Hdider C., Lenuci M.S. When Color Really Matters: Horticultural Performance and Functional Quality of High-Lycopene Tomatoes. Crit Rev Plant Sci, 2018, 37(1), 15-53. doi.org/10.1080/07352689.2018.1465631
Line 70, 72 addressed
Line 82 – used abbreviation for ET
Line 93-94 addressed: …owing to a higher (10.5%) number of fruits per plant and the higher mean weight per fruit.
Line 104 add. although the difference was not significant.
Line 114-116 paragraph was deleted
Line 130-134 was add: Significant differences were also recorded between S Ch 300 and IB Ch 200, with a significantly lower number of fruits of cultivar IB chemically fertilized and irrigated according to the 67% ET.
Line 143 previous was deleted
Line 145-146 was change with The lowest yield recorded for the cultivar IB because it is not so well adapted for greenhouse, where the temperatures is higher and air humidity is lower.
Line 197 was deleted that recorded in
Paragraph 198-203 was move above.
Paragraph 211-213 was move above.
Line 227-229 was changed with Regarding the fertilization type, the Ch treatment showed a P concentration in the fruits statistically higher than the Og fertilization
Line 231 was changed with The cultivar Siriana F1 showed a higher Ca value than IB and HTP F1.
Line 265-271 was changed with In our study, the Ca concentration was 2-3 times lowest in the fruits of cultivar HTP F1 under organic fertilization for 200 m3·ha-1 and 300 m3·ha-1 irrigation regime than Ch treatment.
Line 303 Table 5 was changed Table 8
Line 296-298 was deleted
Line 338 added references
Line 361 was add …the experimental surface area was 3.6 m2 (0.30 m2 nutritive surface for each plant), including 12 tomato plants, planted on beds with 1 m between rows and 0.3 m between plants per row.
Line 360 and 382 was add ET, only abbreviation
Line 383 was add: The three cultivars of Lycopersicon esculentum were compared:
Line 42 the extract methodology is better specified in the reference cited

Round 3
Reviewer 1 Report
Report on Plants 872400 v3
Much improved, both with my comments and apparently others’ too.
A few issues to address:
Most of the times you use ‘it was observed that’ or ‘Results showed that’ these phrases are redundant and can/should be removed.
Line 28 to read: concentration of any of the macroelements
Line 113 to read: generally caused a decline in fruit weight,
Line 189/190 to read: Within each column and within a treatment different letters mean treatments differ significantly according to Tukey test at p≤0.05.
Line 199 to read: of the main effects of cultivar, fertilization type
Line 200 to read: that the cultivar had the major effect on the
Line 217 remove this: The differences registered were about 69% (Table 4).
Line 227 to read: the highest P
Line 229 to read: fertilization were significant. Ca was significantly
Line 234 to read: concentration was significantly
Line 251 to read: 74.7% over cultivar
Line 258 to read: found in the highest concentration in the
Line 260 to read: (Table 6). However, the highest
Line 273 to read: Indeed, the mineral concentration is closely
Line 289 to read: Summarising, Cu was influenced
Line 295 to read: Within each column and within a treatment different letters mean treatments differ significantly according to Tukey test at p≤0.05.
Line 322 to read: In other research [40] the
Line 360 to read: 3.6 m2 (0.96 m x 0.25 m),
Line 439-453 are not really Conclusions, rather a [poor] summary of the results…
So, I suggest to read:
Tomato yield and the number of fruits were significantly influenced by cultivar but not by fertilization treatments. The treatment with the greater supply of water led to higher fruit yield. Indeed, highest yield was achieved in the combination of Cultivar HTP F1 under organic fertilization and 100% evapotranspiration replenishment.
That concentrations of phosphorus, calcium, nickel and lead in the fruits of cultivar Siriana F1 were higher under the 67% evapotranspiration replenishment and chemical fertilization, suggests that these mineral elements are more easily absorbed by this cultivar as compared with HTP F1 and Inima de Bou cultivars.
Macroelement concentration of Mg and K in tomato fruits were affected by the fertilization type, P by cultivar and fertilization type and Ca by all the factors examined. Microelement concentrations in tomato fruits were influenced by treatments. Copper was affected only by the fertilization type, Mn and Ni by cultivar and irrigation regime, and Pb by all three factors. The positive effect of the organic fertilization on the lycopene and antioxidant activity of tomato fruits showed that this fertilization type can be used as an alternative to the chemical fertilization in sustainable agriculture, giving premium quality products.
Author Response
Reviewer 1
Most of the times you use ‘it was observed that’ or ‘Results showed that’ these phrases are redundant and can/should be removed
Comment: “The results showed” addressed for lines 95, 211, 231
Line 28 to read: concentration of any of the macroelements
Comment: was addressed
Line 113 to read: generally caused a decline in fruit weight
Comment: was addressed
Line 189/190 to read: Within each column and within a treatment different letters mean treatments differ significantly according to Tukey test at p≤0.05
Comment: was addressed
Line 199 to read: of the main effects of cultivar, fertilization type
Comment: was addressed with Regarding of the main effects of cultivar, fertilization type
Line 200 to read: that the cultivar had the major effect on the
Comment: addressed
Line 217 remove this: The differences registered were about 69% (Table 4).
Comment: this phrase was removed
Line 227 to read: the highest P
Comment: was addressed
Line 229 to read: fertilization were significant. Ca was significantly
Comment: addressed
Line 234 to read: concentration was significantly
Comment: Addressed
Line 251 to read: 74.7% over cultivar
Comment: addressed
Line 258 to read: found in the highest concentration in the
Comment: amount was change with concentration
Line 260 to read: (Table 6). However, the highest
Comment: anyway was change with however
Line 273 to read: Indeed, the mineral concentration is closely
Comment: Content was change with concentration
Line 289 to read: Summarising, Cu was influenced
Comment: Was addressed
Line 295 to read: Within each column and within a treatment different letters mean treatments differ significantly according to Tukey test at p≤0.05.
Comment: was addressed
Line 322 to read: In other research [40] the
Comment: was addressed
Line 360 to read: 3.6 m2 (0.96 m x 0.25 m)
Comment: was addressed
Line 439-453 are not really Conclusions, rather a [poor] summary of the results…
So, I suggest to read:
Tomato yield and the number of fruits were significantly influenced by cultivar but not by fertilization treatments. The treatment with the greater supply of water led to higher fruit yield. Indeed, highest yield was achieved in the combination of Cultivar HTP F1 under organic fertilization and 100% evapotranspiration replenishment.
That concentrations of phosphorus, calcium, nickel and lead in the fruits of cultivar Siriana F1 were higher under the 67% evapotranspiration replenishment and chemical fertilization, suggests that these mineral elements are more easily absorbed by this cultivar as compared with HTP F1 and Inima de Bou cultivars.
Macroelement concentration of Mg and K in tomato fruits were affected by the fertilization type, P by cultivar and fertilization type and Ca by all the factors examined. Microelement concentrations in tomato fruits were influenced by treatments. Copper was affected only by the fertilization type, Mn and Ni by cultivar and irrigation regime, and Pb by all three factors. The positive effect of the organic fertilization on the lycopene and antioxidant activity of tomato fruits showed that this fertilization type can be used as an alternative to the chemical fertilization in sustainable agriculture, giving premium quality products.
Comment: thank you very much for conclusions suggested. Are much better now.
Tomato yield and the number of fruits were significantly influenced by cultivar but not by fertilization treatments. The treatment with the greater supply of water led to higher fruit yield. Indeed, highest yield was achieved in the combination of HTP F1 cultivar under organic fertilization and 100% evapotranspiration replenishment.
That concentrations of phosphorus, calcium, nickel and lead in the fruits of Siriana F1 cultivar were higher under the 67% evapotranspiration replenishment and chemical fertilization, suggests that these mineral elements are more easily absorbed by this cultivar as compared with HTP F1 and Inima de Bou cultivars.
Macroelement concentration of manganese and potasium in tomato fruits were affected by the fertilization type, phosphorus by cultivar and fertilization type and calcium by all the factors examined. Microelement concentrations in tomato fruits were influenced by treatments. Copper was affected only by the fertilization type, manganese and nickel by cultivar and irrigation regime, and lead by all three factors. The positive effect of the organic fertilization on the lycopene and antioxidant activity of tomato fruits showed that this fertilization type can be used as an alternative to the chemical fertilization in sustainable agriculture, giving premium quality products.

This manuscript is a resubmission of an earlier submission. The following is a list of the peer review reports and author responses from that submission.
Round 1
Reviewer 1 Report
The manuscript contains numerous errors in setting up and writing. Here are some suggestions for improving the manuscript.
Bibliographic references 1, 2, 5, 6, 7, 9, 10, 15, 33, 53 not pertinent to what has been stated
lines 53-55: sentence and references not consistent with the scientific starting point of the manuscript
lines 58-59: clarify this part of the phrase "and 5-10 / 5 of the daily dose of P and Mg"
Lines 68, 70, 77, 314, 330: clarify the abbreviations d.w., f.w., GAE, Og, BBCH 805–808
The introduction deals more with the quality of the tomato than the cultivation techniques (irrigation and fertilization) on which the experimental work was set
Line 301: what was the arrangement of factors in the experimental design?
Lines 303-304: indicate the manufacturer of the seeds and the type of fruit of the three varieties.
Lines 309-310: the ratio between the quantities of fertilizer used is not equal to the ratio between the percentages of nitrogen in the two fertilizers (and the explanation given in lines 313-316 is not acceptable).
Lines 317-319: "Tomato plants were irrigated 26 times over the whole crop season (as an average of the two crop seasons) according to two different water volumes, based on 67% or 100% of evapotranspiration replenishment: 200 m3·ha−1 (5.200 m3 overall) and 300 m3·ha−1 (7.800 m3 overall) at each irrigation." Use this (67% or 100% of evapotranspiration replenishment) and not 200 and 300 m3 in the abstract and results.
line 338: indicate with a number the reference Butnariu and Giuchici, 2011
lines 339-341, 365, 371: change ml to mL
lines 381-382: The data were statistically processed by two-way ANOVA or three-way ANOVA?
Why are there no references to the two years of experimentation in the results?
Table 1 does not indicate the significance of two treatments and for two dependent variables ... The unit of measurement of the number of fruits (n / plant?) Is missing. In table 2 the result of the interactions is missing (the three-way interaction would also be missing).
It is not correct to report in the results the values already present in the tables (see lines 89-90, 95, 96, 99-100, 106, etc)
Lines 90-91: This statement ("This indicates that the biometric parameters are presumably influenced by genotype and not by the treatment applied") cannot be derived from table 1 (but from any interactions).
Lines 98-100: the claim is not demonstrated by ANOVA
Lines 118-120: the data in table 2 is reported for a variable that was not significantly influenced by the interactions (which ones?)
Author Response
Dear Reviewer, we wish to thank you for contributing to improve our manuscript.
Reviewer 1
Comments and Suggestions for Authors
The manuscript contains numerous errors in setting up and writing. Here are some suggestions for improving the manuscript.
Bibliographic references 1, 2, 5, 6, 7, 9, 10, 15, 33, 53 not pertinent to what has been stated
Dear Reviewer, we have addressed all your above recommendations, except for the number 9 citation, as it can be checked in the References section.
lines 53-55: sentence and references not consistent with the scientific starting point of the manuscript
Addressed (L 59-61).
lines 58-59: clarify this part of the phrase "and 5-10 / 5 of the daily dose of P and Mg"
We have clarified the above mentioned sentence (L 64-65).
Lines 68, 70, 77, 314, 330: clarify the abbreviations d.w., f.w., GAE, Og, BBCH 805–808
We have clarified the above abbreviations.
The introduction deals more with the quality of the tomato than the cultivation techniques (irrigation and fertilization) on which the experimental work was set
We have added three citations relevant to the cultivation techniques (L 51-56).
Line 301: what was the arrangement of factors in the experimental design?
We have detailed the experimental design as required (L 276-277).
Lines 303-304: indicate the manufacturer of the seeds and the type of fruit of the three varieties.
We have reported the information requested (L 300-302).
Lines 309-310: the ratio between the quantities of fertilizer used is not equal to the ratio between the percentages of nitrogen in the two fertilizers (and the explanation given in lines 313-316 is not acceptable).
We have further detailed the explanation relevant to the fertilization choices (L 295-299).
Lines 317-319: "Tomato plants were irrigated 26 times over the whole crop season (as an average of the two crop seasons) according to two different water volumes, based on 67% or 100% of evapotranspiration replenishment: 200 m3·ha−1 (5.200 m3 overall) and 300 m3·ha−1 (7.800 m3 overall) at each irrigation." Use this (67% or 100% of evapotranspiration replenishment) and not 200 and 300 m3 in the abstract and results.
Addressed (L 22-23).
line 338: indicate with a number the reference Butnariu and Giuchici, 2011
We have added the citation number (L 323).
lines 339-341, 365, 371: change ml to mL
Addressed.
lines 381-382: The data were statistically processed by two-way ANOVA or three-way ANOVA?
We have clarified the above information (L 353).
Why are there no references to the two years of experimentation in the results?
We have added a sentence at the beginning of the Results section (L 92-95).
Table 1 does not indicate the significance of two treatments and for two dependent variables ... The unit of measurement of the number of fruits (n / plant?) Is missing. In table 2 the result of the interactions is missing (the three-way interaction would also be missing).
We have completed the Table with the recommended information.
It is not correct to report in the results the values already present in the tables (see lines 89-90, 95, 96, 99-100, 106, etc)
We have replaced all the mentioned values with percentage values relevant to the comparisons.
Lines 90-91: This statement ("This indicates that the biometric parameters are presumably influenced by genotype and not by the treatment applied") cannot be derived from table 1 (but from any interactions).
We have modified the sentence (L 101-106).
Lines 98-100: the claim is not demonstrated by ANOVA
We have amended the sentence (L 107-110).
Lines 118-120: the data in table 2 is reported for a variable that was not significantly influenced by the interactions (which ones?)
We have addressed this comment (L127-128).
Dear Reviewer, we wish to thank you for contributing to improve our manuscript.
Reviewer 2
Comments and Suggestions for Authors
Various technological methods of tomato cultivation are known, their qualitative and quantitative indicators depending on various biotic and abiotic indicators. Our workplace also deals with this. The presented work is valuable in terms of the impact of varietal diversity and irrigation regime on qualitative and quantitative parameters, so I recommend its publication.
My recommendations are:
Can you add whether the monitored risk elements (Pb, Cr, Ni, Zn, Cu) did not exceed the permitted limit of the maximum permissible amount in the fruit?
We have added the above information (L 253-254).
- line 330-331: What was the weight of the samples collected? Were they the same from each plot?
We have reported the detailed required (L 314-315).
- line 338: How many samples were homogenized?
- line 346: Fill in the value of E
In response to the two above questions, we have shortened the method description as recommended by another Reviewer (L 222-223).
- line 349: What was the weight of the fruit in the 70% ethanol extract?
Addressed as above mentioned.
Dear Reviewer, we wish to thank you for contributing to improve our manuscript.
Reviewer 3
Comments and Suggestions for Authors
Report on Plants 821179
Much effort has gone into the design and running of this [commendable] two-year experiment investigating varietal response to irrigation amounts and to chemical vs. organic fertilization. The data appear to be reliable, but there are many detractions that make the manuscript in its current form unpublishable.
I have heavily annotated the manuscript and copied into a pdf file, and the authors should take note of all that I mention, to improve the presentation so that it may in the future be acceptable for publication. In the results it seems as though the authors wish to write about everything they found, but they must be more objective and only report what is interesting, relevant and significant..
Also write about ‘weight per fruit’ and not ‘fruit weight’. And concentration, not content [for content = concentration * mass].
One of the main issues, is with the expression in English, another is the lack of understanding of what and how interactions work, and the third is to try to present too much information. Ideally the results and discussion section would be halved in length, as will the number of references. The later are not all justified, and the same reference can be used to substantiate various points made. Indeed, on a number of occasions, I find that the reference cited is only obtusely [if at all] related to the point being made. I will indicate some instances of this in the text and as follows:
We have addressed all the comments highlighted inside the ‘pdf’ text.
Line 69 references 30 and 31 do not refer to lycopene concentrations as affected by types of fertilizer. In fact reference 30 does not mention lycopene.
We have deleted the reference 30.
Line 72 does not mention sucrose, and is 0.02 g/100 g f.w. a high value for sucrose??!
We have modified the sentence (L 77).
Line 76 reference 33 has no mention of organic production, and the values reported in the current text are for skin and pulp!!
We have deleted the reference 33.
Line 167 reference 43 [only 9 variety comparison] is not relevant, and reference 45 relates to microbial cultures, not to the field.
Line 190 reference 43 [only 9 variety comparison] is not relevant
We have deleted the above mentioned references.
Line 272/273 I see values reported of 180 ppm, how reconcile with what you write?
Addressed as a compromise of all the Reviewers’ recommendations (L 238-241).
Line 330 put (BBCH 805–808) in full.
We deleted the mentioned expression.
Line 436 and elsewhere, remove Basel
Addressed.
Line 552 this section is very similar to reference 54..
Addressed.

Reviewer 2 Report
Various technological methods of tomato cultivation are known, their qualitative and quantitative indicators depending on various biotic and abiotic indicators. Our workplace also deals with this. The presented work is valuable in terms of the impact of varietal diversity and irrigation regime on qualitative and quantitative parameters, so I recommend its publication.
My recommendations are:
Can you add whether the monitored risk elements (Pb, Cr, Ni, Zn, Cu) did not exceed the permitted limit of the maximum permissible amount in the fruit?
- line 330-331: What was the weight of the samples collected? Were they the same from each plot?
- line 338: How many samples were homogenized?
- line 346: Fill in the value of E
- line 349: What was the weight of the fruit in the 70% ethanol extract?
Author Response

(The authors gave the same response as above.)

Reviewer 3 Report
Report on Plants 821179
Much effort has gone into the design and running of this [commendable] two-year experiment investigating varietal response to irrigation amounts and to chemical vs. organic fertilization. The data appear to be reliable, but there are many detractions that make the manuscript in its current form unpublishable.
I have heavily annotated the manuscript and copied into a pdf file, and the authors should take note of all that I mention, to improve the presentation so that it may in the future be acceptable for publication. In the results it seems as though the authors wish to write about everything they found, but they must be more objective and only report what is interesting, relevant and significant..
Also write about ‘weight per fruit’ and not ‘fruit weight’. And concentration, not content [for content = concentration * mass].
One of the main issues, is with the expression in English, another is the lack of understanding of what and how interactions work, and the third is to try to present too much information. Ideally the results and discussion section would be halved in length, as will the number of references. The later are not all justified, and the same reference can be used to substantiate various points made. Indeed, on a number of occasions, I find that the reference cited is only obtusely [if at all] related to the point being made. I will indicate some instances of this in the text and as follows:
Line 69 references 30 and 31 do not refer to lycopene concentrations as affected by types of fertilizer. In fact reference 30 does not mention lycopene.
Line 72 does not mention sucrose, and is 0.02 g/100 g f.w. a high value for sucrose??!
Line 76 reference 33 has no mention of organic production, and the values reported in the current text are for skin and pulp!!
Lin 167 reference 43 [only 9 variety comparison] is not relevant, and reference 45 relates to microbial cultures, not to the field.
Line 190 reference 43 [only 9 variety comparison] is not relevant
Line 272/273 I see values reported of 180 ppm, how reconcile with what you write?
Line 330 put (BBCH 805–808) in full.
Line 436 and elsewhere, remove Basel
Line 552 this section is very similar to reference 54..

Author Response

(The authors gave the same response as above.)

Round 2
Reviewer 3 Report
Report on v2 Plants 821179
This is an improved version of the manuscript, but still not publishable.
The authors have not paid attention to all of my requested changes, and have not indicated why [for example, ‘weight per fruit’ is not used, nor is ‘concentration’, and they must be].
Again, I have annotated the text.
Major issues relate to the convoluted expression in English and sentences or phrases with no sense [and it is not my job to rewrite the text], to the lack of data [a major omission] on main effects of treatments on lycopene, polyphenol content and antioxidant activity and on macro and micro-nutrient parameters, and a few other issues, such as the very low SEs in Table 1 for mean weight per fruit but lack of treatment effects, some lack of consistency between the text and data presented [e.g., lines 132-134 and Table 1; lines 174-175 and Table 3..], and really no conclusions in the Conclusions section other than the last sentence.
